# Experiences of Korean Medicine treatment in patients with Bell's palsy: A qualitative study protocol

**Eunbyul Cho** [1©¤], **Seung Eun Chung** [2©], **Sungha Kim** [1]*

1 Korean Medicine Science Research Division, Korea Institute of Oriental Medicine, Daejeon, Republic of Korea, 2 Department of Nursing, Korea National University of Transportation, Jeungpyeong-gun, Republic of Korea

© These authors contributed equally to this work.
¤ Current Address: Department of Diagnostics, College of Korean Medicine, Wonkwang University, Iksan, Republic of Korea
* bozzol@kiom.re.kr

## Abstract

### Background

Bell's palsy significantly impacts patients' quality of life, with approximately 30% not fully recovering. In South Korea, Korean Medicine (KM) is widely used as a complementary approach for facial palsy, with 50.86% of patients utilizing KM in 2021. Although quantitative studies have shown KM's effectiveness, there is a lack of research on lived experiences of the participants on KM treatment. We aim to identify the meaning of using KM among the patients with Bell's palsy.

### Methods

This study will be conducted in compliance with the consolidated criteria for reporting qualitative research. Patients with Bell's palsy who had or have received KM treatment and whose onset was less than two years prior to the interview date will be recruited by purposive and snowball sampling. One-on-one interviews will be conducted in person using a semi-structured interview guide. Interviews and recruitment will continue until meaning saturation is reached. The interviews will be conducted in South Korea in a private area near the patient's residence to make them feel comfortable. The data will be analyzed using Colaizzi's seven-step method.

### Discussion

This study will explore the lived experiences of patients using KM for Bell's palsy and seek to understand the essence of their experiences. Utilizing Colaizzi's phenomenological approach, common themes in patients' experiences of receiving KM treatment

**Data availability statement:** No datasets were generated or analysed during the current study. All relevant data from this study will be made available upon study completion. All data regarding the present protocol are in the manuscript and/or supporting information files.

**Funding:** This work was supported by the Korea Health Technology R&D Project through the Korea Health Industry Development Institute (RS-2022-KH127692). The funder had no role in study design, decision to publish, or preparation of the manuscript.

**Competing interests:** There are no conflicts of interest to declare. This does not alter our adherence to PLOS ONE policies on sharing data and materials.

will be uncovered. The findings are expected to provide valuable insights into healthcare professionals, potentially enhancing clinical practice and patient-centered care in the treatment of Bell's palsy.

## Introduction

Bell's palsy, the most prevalent diagnosis of facial nerve weakness or paralysis [1], greatly impacts the physical, psychological, and social functioning of those who have this condition [2]. With approximately 30% of those diagnosed with Bell's palsy not fully recovering [1], patients desire continuous care to improve facial function and manage complications, such as facial asymmetry, synkinesis, and hypertonia [3]. Current clinical-practice guidelines recommend oral steroids within 72 hours of symptom onset [4]. However, treatment dissatisfaction combined with limited therapeutic options in conventional medicine (CM) has created significant demand for additional therapies such as herbal medicine and acupuncture [5–7].

In South Korea, where the healthcare system uniquely allows patients access to diverse treatment options, Korean Medicine (KM) is often used as a complementary approach alongside CM for treating Bell's palsy [8,9]. KM offers various modalities for Bell's palsy, such as acupuncture with electric stimulation, herbal medicine, pharmacopuncture (injection of herbal extracts into acupuncture points), thread-embedding acupuncture (insertion of absorbable threads), moxibustion (burning of mugwort herb on or near the skin), and cupping (suction therapy using cups) [10]. The effectiveness of these treatments is supported by the recent clinical-practice guideline of KM for Bell's palsy [11]. A meta-analysis of randomized controlled trials revealed that acupuncture combined with oral steroids was more effective than steroid monotherapy [11]. Another meta-analysis showed a significant difference favoring the combination of herbal medicine Gyeonjeong-san with Western medication (WM) over WM alone for improving facial symptoms.

As a result of this growing evidence and patients' needs for active treatment, the utilization of KM reached 50.86% among patients with facial palsy in South Korea who sought medical care in 2021 [8]. Despite this widespread use, there is a notable lack of research on patients' subjective experiences during treatment. These experiences are closely associated with patient satisfaction and quality of life [12]. Understanding patient experiences is crucial for identifying areas of improvement and providing valuable insights to healthcare professionals treating Bell's palsy. Previous studies have primarily focused on physical, psychological, and social alterations following facial palsy [2,5,13]. One qualitative interview with Korean Bell's palsy patients limited its scope to hospitalization experiences [6]. Another qualitative study applied grounded theory to focus on the process of choosing KM after CM treatment, but it mixed various etiologies of facial palsy—including cholesteatoma, trauma, Ramsay Hunt syndrome, and central facial palsy—thus lacking a dedicated exploration of Bell's palsy–specific patient experiences [7]. Therefore, further descriptive study is required to gain insights into patients' experiences with KM treatment for Bell's palsy and to understand the meaning of using KM.

The present study will employ a descriptive phenomenological approach to explore the lived experiences of patients with Bell's palsy who received KM treatment. We aim to uncover the essence structure of this phenomenon. We will examine patients' motivations for seeking KM, their perceptions and experiences during treatment, and the perceived impact on their recovery journey.

## Materials and methods

### Study design

This qualitative study will be conducted by doctors of Korean Medicine (KMDs) and a nurse. It will employ a phenomenological methodology to understand the meaning of experiences related to the use of KM treatment for Bell's palsy. We will explore lived experiences of the participants through in-depth interviews to reveal the essence structure of the phenomenon. This study will be conducted following the consolidated criteria for reporting qualitative research (COREQ) (S1 Checklist) [14]. We provide an English version of the full protocol (S2 File).

### Inclusion criteria

This study will include patients with Bell's palsy who developed the condition less than two years prior to the interview date. The period of the illness is limited to avoid recall bias; a previous study has found that interviews are less accurate for experiences older than two years [15]. The age criterion will be being aged above 19, which is the legal threshold for adulthood in South Korea. Patients who receive at least one KM treatment, regardless of the type of treatment, will be included. We will exclude individuals who have declined to participate in the study, those who have difficulties with verbal communication or cognitive functioning, and those deemed unsuitable by the study's administrator. Factors such as age, sex, duration of KM treatment, and Bell's palsy prognosis will be considered to obtain diverse patient experiences.

### Sample size

The concept of saturation is key in determining the final sample size for qualitative research. Saturation refers to the point at which no additional meaningful data can be obtained from new participants [16]. In this study, interviews will continue until saturation is achieved. We have set an initial target sample size for 30 participants, but this number may be adjusted based on when saturation is reached. If saturation is not achieved with 30 participants, additional interviews will be conducted; conversely, if saturation is reached earlier, data collection may conclude with fewer participants.

### Participant recruitment

We will use purposive sampling to recruit participants from institutions recommended by Korean Acupuncture & Moxibustion Medicine Society and the Society of KM Ophthalmology, Otolaryngology & Dermatology, which are closely related to facial palsy. Individuals who are willing to take part in the study may contact the researcher. Following the initial interview, each participant will be asked to suggest candidates for further interviews. If a potential participant expresses an interest in taking part in the study, the researcher will explain the study to them and ask for their intention to participate. Participant recruitment began on July 5, 2024.

A potential limitation of our recruitment methods is the risk of selection bias. Purposive sampling through KM-related professional societies may result in referrals of patients who had positive treatment experiences, as practitioners are more likely to recommend satisfied patients. Snowball sampling may result in participants with similar experiences or attitudes towards KM treatment. To address this bias, we will actively encourage referring KMDs and participants to include individuals with varied treatment outcomes, including those with neutral or negative experiences. We will also monitor recruitment patterns to ensure diversity in participant demographics and experiences, including variations in age, sex, treatment duration, and Bell's palsy prognosis.

 

## Data collection

The interviews are scheduled for the period between July and December 2024. We will collect data through semi-structured interviews to enable flexibility, explore new ideas, and enhance in-depth understanding [17]. Face-to-face interviews will be conducted to allow the participants to fully express their experiences. All the interviews will be conducted in Korean, with each interview lasting 60–90 minutes. Each participant may be interviewed up to three times if needed.

The interviews will be conducted near the participant's residence in a quiet venue, such as a small meeting room, to ensure participant comfort. If an in-person interview is not possible or the participant feels uncomfortable, online interviews will be conducted using Zoom. During the interviews, the interviewer will observe non-verbal communication, including voice, facial expressions, and gestures. Interviews and recruitment will continue until saturation is reached. Each participant will be compensated with KRW 100,000 per interview. To minimize potential response bias related to compensation, participants will be clearly informed that honest and authentic sharing of experiences, whether positive, negative, or neutral, is valued equally and will not affect their compensation.

The interview guide has been developed based on Colaizzi's method [18] and informed by previous qualitative studies on facial palsy patients [5–7,13]. The initial draft was then reviewed and refined through consultations with an external qualitative research expert to ensure its relevance. Each participant will be asked about their sex, age, region of current residence, occupation, date of onset, current treatment for Bell's palsy, current medications for underlying medical conditions, and types of KM treatment received to date. The interview guide covers two main domains: (1) Background information and treatment history, (2) Experience with KM treatment for Bell's palsy. Core questions include "Please tell me your overall impression (feeling) when you think about your KM treatment," and "Did you notice any changes in your facial condition after you received the treatment?" (S2 File).

The participants will be encouraged to freely express their experiences. To minimize social desirability bias and encourage authentic responses, participants will be continuously reminded that sharing challenges or disappointments with treatment is equally important as positive outcomes for the research. Participants will be assured of confidentiality, with explicit guarantees that their responses will remain anonymous and will not be shared with their treating KMDs or any other parties [19].

Interviews and analysis will be conducted by three experienced female researchers. Two of them are KMDs, have PhD in KM, and are specialists in acupuncture and moxibustion, which are the most commonly used treatments for Bell's palsy. The third interviewer, a nurse with a PhD in nursing, is an expert in qualitative research who has over 30 years of experience, has served as the president of the Academy of Qualitative Research, and has presented numerous qualitative studies at conferences related to health and nursing. The interviewers will suspend judgment to ensure that their opinions do not influence the data collection process. To ensure reflexivity and minimize potential bias arising from the researchers' clinical background, all interviewers will maintain field notes throughout the data collection process to document their assumptions, reactions, and potential biases encountered during each interview.

Recognizing that discussing illness experiences may provoke emotional distress, the research team—which includes healthcare professionals—will monitor participant wellbeing throughout interviews. Participants will be informed of their right to pause or stop the interview at any time.

## Data analysis

All interviews will be audio-recorded and transcribed verbatim. The data will be analyzed using Colaizzi's seven-step phenomenological method of analysis [18]. The following procedures will be implemented. First, the researchers will read through all the participants' interviews several times to familiarize themselves with the data. Second, they will identify significant statements that are of direct relevance to the phenomenon under investigation. Third, the researchers will formulate meanings from the meaningful statements, while consistently bracketing their pre-suppositions about the

phenomenon. Fourth, they will cluster the identified meanings into themes that are universally present in all the narratives. Fifth, a comprehensive and inclusive description of the phenomenon will be written, including all the themes produced. Sixth, the researchers will write a dense but clear statement (i.e., an exhaustive description) of the phenomenon, which is essential for explaining its structure. Seventh, they will present the fundamental structure statement to the participants to check whether it accurately represents their experiences. Based on the feedback from the participants, the researchers may go back to the previous steps and modify the analysis [20].

To ensure the reliability and validity of the qualitative research, triangulation will be implemented to cross-validate the findings. Data triangulation will be accomplished by consulting multiple sources, including the interview transcripts and researchers' field notes, and by analyzing them together. Researcher triangulation will also be conducted. Additionally, reflexive memos will be written during the analysis phase to record researcher thoughts and interpretations while coding participant narratives. The qualitative research expert (nurse researcher) will serve as an external monitor, challenging assumptions made by the KMD researchers and ensuring methodological rigor throughout the analysis process. When the primary interviewer conducts the interviews and analysis, the principal investigator and qualitative-research expert will consult with the primary interviewer and supervise the research process to ensure that bias is minimized. Qualitative data analysis software ATLAS.ti for Windows will be used to maintain consistency in the coding and data results. The findings will be validated by returning them to the research subjects to obtain their feedback [18].

## Ethical considerations

This study was reviewed and approved by the institutional review board of the Korea Institute of Oriental Medicine (I-2404/004-003-02). Prior to the interviews, participants will be provided with detailed information regarding the purpose of the interview, the protection of their personal information and privacy, any potential risks and benefits associated with participating, and their right to withdraw if desired.

Additionally, participants will be informed about the researchers' backgrounds as KMDs and qualitative research experts. The researchers will explain their professional interest in understanding patients' experiences with KM for Bell's palsy and how this research aims to improve patient care. This is to ensure confidentiality and guarantee that the subjects are fully informed and take part voluntarily. The participants will provide their informed consent by signing a document. They will also sign a separate document that grants permission for the interviews to be recorded. To protect the participants' privacy, their identities will be anonymized through the allocation of codes. Personal information will be collected to reward participants for taking part in the study, but it will be kept confidential. All the documents relating to this study will be retained for three years following its conclusion; afterward, they will be erased according to the Personal Information Protection Act of South Korea.

## Discussion

This study aims to explore the lived experience of patients with Bell's palsy who had or have undergone KM treatment in South Korea. Despite the widespread use and growing evidence of KM for Bell's palsy in South Korea, there is a need for further in-depth investigation to explore the subjective experiences of patients. By examining patients' motivations, perceptions, and the perceived impact of KM treatment on their recovery journey, this research will attempt to provide valuable insights that can inform clinical practice and enhance patient-centered care.

This study will use a phenomenological methodology, which involves analyzing individuals' descriptions of their lived experiences to clarify the meanings present in them [21]. In particular, Colaizzi's method focuses on the identification of shared characteristics among participants, rather than individual characteristics [20]. While the participants will differ in terms of gender, age, and Bell's palsy prognosis, the study will shed light on their common experiences of receiving KM treatment and their perceptions of it.

This study will be the first phenomenological qualitative research to explore the essential structure of KM use for Bell's palsy. The findings will suggest implications for not only KMDs but also healthcare providers worldwide who are interested in integrative approaches for treating Bell's palsy. Nevertheless, the generalizability of findings to other contexts may be limited by South Korea's distinctive dual healthcare system that allows concurrent access to both KM and CM, as well as cultural factors that may influence patient attitudes toward complementary therapies.

## Dissemination

We intend to disseminate the findings of this study through academic journals, seminars, and conference presentations, including the preparation of multilingual abstracts to reach international audiences. The study's target audience comprises healthcare providers, including KMDs, as well as patients with Bell's palsy. Furthermore, we plan to propose the results at the health policy development meeting. Therefore, the findings can be disseminated in academics, policy, and clinical practice.

## Supporting information

**S1 Checklist. COREQ checklist.**
(PDF)

**S1 File. English version of study protocol for Institutional Review Board.**
(PDF)

**S2 File. Interview guide.**
(PDF)

## Author contributions

**Conceptualization:** Eunbyul Cho, Seung Eun Chung, Sungha Kim.

**Funding acquisition:** Sungha Kim.

**Investigation:** Eunbyul Cho, Seung Eun Chung, Sungha Kim.

**Methodology:** Eunbyul Cho, Seung Eun Chung, Sungha Kim.

**Project administration:** Sungha Kim.

**Supervision:** Seung Eun Chung, Sungha Kim.

**Writing – original draft:** Eunbyul Cho.

**Writing – review & editing:** Eunbyul Cho, Seung Eun Chung, Sungha Kim.

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
