## [Decision Letter · Decision Letter 0]

3 Jul 2025

Dear Dr. Kim,

Thank you for submitting your manuscript to PLOS ONE. After careful consideration, we feel that it has merit but does not fully meet PLOS ONE’s publication criteria as it currently stands. Therefore, we invite you to submit a revised version of the manuscript that addresses the points raised during the review process.

We look forward to receiving your revised manuscript.

Kind regards,

Jaspinder Kaur, MD

Academic Editor

PLOS ONE

Journal Requirements:

The authors declare no conflict of interests.

5. Please remove all personal information, ensure that the data shared are in accordance with participant consent, and re-upload a fully anonymized data set.

Additional guidance on preparing raw data for publication can be found in our Data Policy (https://journals.plos.org/plosone/s/data-availability#loc-human-research-participant-data-and-other-sensitive-data) and in the following article: http://www.bmj.com/content/340/bmj.c181.long .

Reviewers' comments:

Reviewer's Responses to Questions

**Comments to the Author**

1. Does the manuscript provide a valid rationale for the proposed study, with clearly identified and justified research questions?

Reviewer #1: Yes

Reviewer #2: Yes

2. Is the protocol technically sound and planned in a manner that will lead to a meaningful outcome and allow testing the stated hypotheses?

Reviewer #1: Yes

Reviewer #2: Yes

3. Is the methodology feasible and described in sufficient detail to allow the work to be replicable?

Reviewer #1: Yes

Reviewer #2: Yes

4. Have the authors described where all data underlying the findings will be made available when the study is complete?

Reviewer #1: Yes

Reviewer #2: No

5. Is the manuscript presented in an intelligible fashion and written in standard English?

Reviewer #1: Yes

Reviewer #2: Yes

You may also provide optional suggestions and comments to authors that they might find helpful in planning their study.

Reviewer #1: This protocol outlines a timely and relevant qualitative study aiming to explore patients' lived experiences with Korean Medicine (KM) in treating Bell’s palsy. The rationale is clearly established, highlighting the growing use of KM and the existing gap in literature regarding patient perspectives.

The research question is well justified and aligns with a meaningful knowledge gap in both complementary medicine and patient-centered care domains. The study design using Colaizzi’s phenomenological approach is appropriate for addressing the research aims, and the inclusion of semi-structured interviews with purposive and snowball sampling ensures a diverse and contextually rich sample.

The methodological details are thorough and transparent. The inclusion criteria, sample size considerations (based on saturation), ethical safeguards, interviewer qualifications, and analytical framework are clearly described, enhancing the study’s credibility and reproducibility.

The protocol also thoughtfully addresses potential researcher bias by implementing triangulation and bracketing, and by involving a team with extensive qualitative research experience. The plan for returning findings to participants for validation further strengthens the study's rigor.

Language throughout the manuscript is clear and professionally written. I found no major grammatical or typographical errors. However, consider minor proofreading for fluency, such as rewording overly long sentences (e.g., lines 69–72).

In summary, this is a well-conceived and carefully structured study protocol with potential to yield meaningful insights into patient experiences with KM treatment for Bell’s palsy. I have no major revisions to suggest at this stage.

Reviewer #2: Clarify Interview Guide Domains:

Please consider including the key domains or sample questions from your semi-structured interview guide in the main manuscript (or as a summary table/appendix). This will help readers assess the depth and appropriateness of your planned data collection.

Address Interviewer Reflexivity and Bias:

Since the research team includes KM practitioners and a qualitative research expert, please elaborate on the steps you will take to ensure reflexivity and minimize interviewer bias during both data collection and analysis. For example, will you use reflective journals, debriefing sessions, or other strategies?

Discuss Potential Selection Bias:

Acknowledge and discuss the possibility of selection bias arising from your recruitment methods (purposive sampling through KM-related societies and snowball sampling). Consider whether these strategies may overrepresent participants with positive experiences.

Influence of Financial Compensation:

Please discuss whether offering KRW 100,000 per interview may influence participation and responses. Consider outlining how you will address any resulting bias or participant motivation concerns.

Diversity of Participant Experience:

Clarify how you will ensure a diversity of perspectives (e.g., varying ages, treatment durations, severity/recovery status) among participants to capture a broad range of experiences.

Handling of Negative Experiences and Social Desirability Bias:

Specify how your study will encourage participants to share both positive and negative experiences with KM, and how you will minimize social desirability bias during interviews.

Risk Management for Emotional Distress:

Since discussing illness experiences may provoke distress, outline whether any support or referral mechanisms will be in place for participants who become upset during interviews.

Generalizability of Findings:

Briefly discuss the potential limitations regarding the transferability or generalizability of your findings outside the South Korean context.

Enhance Readability for International Audience:

Provide brief definitions or explanations for KM modalities (such as pharmacopuncture or moxibustion) for readers who may not be familiar with these terms.

Minor Language and Formatting:

Carefully proofread the manuscript for minor grammatical, typographical, and formatting errors to improve clarity and readability.

**Do you want your identity to be public for this peer review?** For information about this choice, including consent withdrawal, please see our Privacy Policy

Reviewer #1: **Yes: ** Shrushti Shah

Reviewer #2: No

---

## [Author Response · Author response to Decision Letter 1]

20 Jul 2025

(Please refer to the attached 'Responses to Reviewers'.)

[Jul 20, 2025]

Dear Editor:

We are pleased to submit a revised version of our manuscript entitled “Experiences of Korean Medicine treatment in patients with Bell’s palsy: a qualitative study protocol” for publication in PLOS ONE.

We thank the reviewers for their constructive comments. Below are our responses to each point:

1. PLOS ONE Style Requirements: We have carefully reviewed our manuscript against the PLOS ONE style templates and made necessary formatting adjustments to ensure compliance with the journal’s requirements. Specifically, we have updated the Supporting Information file naming format to follow PLOS ONE guidelines (e.g., “S1_Checklist”, “S1_File”, “S2_File”).

2. Funding Information: We have corrected the Funding Information (Grant number: RS-2022-KH127692).

As the submission system does not appear to provide a field to revise the Financial Disclosure, we kindly ask if you could replace the existing statement “The author(s) received no specific funding for this work.” with the following:

“This work was supported by the Korea Health Technology R&D Project through the Korea Health Industry Development Institute (RS-2022-KH127692). The funder had no role in study design, decision to publish, or preparation of the manuscript.”

3. Competing Interests: We have updated our competing interests statement in our cover letter as follows: “The authors declare no conflict of interests. This does not alter our adherence to PLOS ONE policies on sharing data and materials.”

4. Data Availability: We have provided the Data Availability Statement in the submission form as recommended: “All data are in the manuscript and/or supporting information files.”

5. Personal Information: As this is a study protocol, no actual participant data has been collected in this paper. We have reviewed all supporting files to ensure no personal information is included.

6. References: We have thoroughly reviewed our reference list and confirmed that all citations are current and accurate, with no retracted papers included.

We appreciate your consideration of our revised manuscript.

Sincerely,

Sungha Kim, KMD, Ph.D.

KM Science Research Division, Korea Institute of Oriental Medicine

(34054) 1672 Yuseong-daero, Yuseong-gu, Daejeon, Republic of Korea

Tel: 82-(0)42-868-9345

E-mail: bozzol@kiom.re.kr

Dear Reviewers,

Thank you for your valuable time and effort in reviewing our manuscript. We believe that the manuscript has been improved based on your thoughtful comments. Changes have been marked as track changes in the manuscript and described with responses.

Reviewer #1

This protocol outlines a timely and relevant qualitative study aiming to explore patients’ lived experiences with Korean Medicine (KM) in treating Bell’s palsy. The rationale is clearly established, highlighting the growing use of KM and the existing gap in literature regarding patient perspectives.

The research question is well justified and aligns with a meaningful knowledge gap in both complementary medicine and patient-centered care domains. The study design using Colaizzi’s phenomenological approach is appropriate for addressing the research aims, and the inclusion of semi-structured interviews with purposive and snowball sampling ensures a diverse and contextually rich sample.

The methodological details are thorough and transparent. The inclusion criteria, sample size considerations (based on saturation), ethical safeguards, interviewer qualifications, and analytical framework are clearly described, enhancing the study’s credibility and reproducibility.

The protocol also thoughtfully addresses potential researcher bias by implementing triangulation and bracketing, and by involving a team with extensive qualitative research experience. The plan for returning findings to participants for validation further strengthens the study's rigor.

Language throughout the manuscript is clear and professionally written. I found no major grammatical or typographical errors. However, consider minor proofreading for fluency, such as rewording overly long sentences (e.g., lines 69–72).

In summary, this is a well-conceived and carefully structured study protocol with potential to yield meaningful insights into patient experiences with KM treatment for Bell’s palsy. I have no major revisions to suggest at this stage.

Response: Thank you for your positive feedback and constructive suggestions. We have revised the manuscript to enhance clarity and readability.

(Page 3, Lines 26–28)

Before revision: However, dissatisfaction with this treatment and the limited therapeutic options of conventional medicine (CM), especially for long-term treatment and management, have led to a significant demand for additional therapies, such as herbal medicine and acupuncture.

After revision: However, treatment dissatisfaction combined with limited therapeutic options in conventional medicine (CM) has created significant demand for additional therapies such as herbal medicine and acupuncture.

(Page 3, Lines 43–46)

Before revision: Despite this widespread use, there is a notable lack of research on patients’ subjective experiences during treatment, which are closely associated with patient satisfaction and quality of life. Understanding these experiences is crucial for identifying areas of improvement and providing valuable insights to health-care professionals treating Bell’s palsy.

After revision: Despite this widespread use, there is a notable lack of research on patients' subjective experiences during treatment. These experiences are closely associated with patient satisfaction and quality of life [12]. Understanding patient experiences is crucial for identifying areas of improvement and providing valuable insights to healthcare professionals treating Bell’s palsy.

(Page 4, Lines 52–56) (Lines 69–72 of the previously submitted manuscript)

Before revision: The present study will employ a descriptive phenomenological approach to explore the lived experiences of patients with Bell’s palsy who had or have undergone KM treatment, seeking to uncover the essence structure of this phenomenon. We will examine patients’ motivations for seeking KM, their perceptions and experiences during treatment, and the perceived impact on their recovery journey.

After revision: The present study will employ a descriptive phenomenological approach to explore the lived experiences of patients with Bell’s palsy who received KM treatment. We aim to uncover the essence structure of this phenomenon. We will examine patients’ motivations for seeking KM, their perceptions and experiences during treatment, and the perceived impact on their recovery journey.

(Page 6, Lines 107–113)

Before revision: All the interviews will be conducted in Korean. Each interview will last 60–90 minutes, and each participant may be interviewed up to three times. The interviews will be conducted near the participant’s residence in a quiet venue, such as a small meeting room, where the participants will feel comfortable. If an in-person interview is not possible or the participant feels uncomfortable, online interviews will be conducted using Zoom. During the interviews, the interviewer will observe non-verbal communication, such as voice, facial expressions, and gestures.

After revision: All interviews will be conducted in Korean, with each interview lasting 60–90 minutes. Each participant may be interviewed up to three times if needed.

The interviews will be conducted near the participant’s residence in a quiet venue, such as a small meeting room, to ensure participant comfort. If an in-person interview is not possible or the participant feels uncomfortable, online interviews will be conducted using Zoom. During the interviews, the interviewer will observe non-verbal communication, including voice, facial expressions, and gestures.

Reviewer #2

[Comment 1] Clarify Interview Guide Domains:

Please consider including the key domains or sample questions from your semi-structured interview guide in the main manuscript (or as a summary table/appendix). This will help readers assess the depth and appropriateness of your planned data collection.

Response: Thank you for your helpful comment. We have added main domains and sample questions of our interview guide in the ‘Data collection section’.

(Page 7, Lines 123–127) The interview guide covers two main domains: (1) Background information and treatment history, (2) Experience with Korean Medicine treatment for Bell’s palsy. Core questions include “Please tell me your overall impression (feeling) when you think about your Korean Medicine treatment,” and “Did you notice any changes in your facial condition after you received the treatment?” (S2 File).

[Comment 2] Address Interviewer Reflexivity and Bias:

Since the research team includes KM practitioners and a qualitative research expert, please elaborate on the steps you will take to ensure reflexivity and minimize interviewer bias during both data collection and analysis. For example, will you use reflective journals, debriefing sessions, or other strategies?

Response: Thank you for your valuable comments. We have added our strategies to ensure reflexivity and minimize interviewer bias in the ‘Data collection’ and ‘Data analysis’ section as follows.

(Page 7, Lines 139–142) To ensure reflexivity and minimize potential bias arising from the researchers’ clinical background, all interviewers will maintain field notes throughout the data collection process to document their assumptions, reactions, and potential biases encountered during each interview.

(Pages 8–9, Lines 164–168) Additionally, reflexive memos will be written during the analysis phase to record researcher thoughts and interpretations while coding participant narratives. The qualitative research expert (nurse researcher) will serve as an external monitor, challenging assumptions made by the KMD researchers and ensuring methodological rigor throughout the analysis process.

[Comment 3] Discuss Potential Selection Bias:

Acknowledge and discuss the possibility of selection bias arising from your recruitment methods (purposive sampling through KM-related societies and snowball sampling). Consider whether these strategies may overrepresent participants with positive experiences.

Response: We sincerely appreciate your comments and suggestions. We have added potential limitations regarding our recruitment methods and our methods to mitigate the selection bias.

(Page 6, Lines 94–101) A potential limitation of our recruitment methods is the risk of selection bias. Purposive sampling through KM-related professional societies may result in referrals of patients who had positive treatment experiences, as practitioners are more likely to recommend satisfied patients. Snowball sampling may result in participants with similar experiences or attitudes towards KM treatment. To address this bias, we will actively encourage referring KMDs and participants to include individuals with varied treatment outcomes, including those with neutral or negative experiences. We will also monitor recruitment patterns to ensure diversity in participant demographics and experiences, including variations in age, sex, treatment duration, and Bell’s palsy prognosis.

[Comment 4] Influence of Financial Compensation:

Please discuss whether offering KRW 100,000 per interview may influence participation and responses. Consider outlining how you will address any resulting bias or participant motivation concerns.

Response: Thank you for your sensitive comments. Considering the resulting bias related to compensation, we have added the following sentence in the ‘Data collection’ section.

(Pages 6–7, Lines 114–117) To minimize potential response bias related to compensation, participants will be clearly informed that honest and authentic sharing of experiences, whether positive, negative, or neutral, is valued equally and will not affect their compensation.

[Comment 5] Diversity of Participant Experience:

Clarify how you will ensure a diversity of perspectives (e.g., varying ages, treatment durations, severity/recovery status) among participants to capture a broad range of experiences.

Response: Thank you for your valuable suggestions. We have added our recruitment strategies to ensure a diversity of perspectives among participants as follows:

(Page 6, Lines 99–101) We will also monitor recruitment patterns to ensure diversity in participant demographics and experiences, including variations in age, sex, treatment duration, and Bell’s palsy prognosis.

[Comment 6] Handling of Negative Experiences and Social Desirability Bias:

Specify how your study will encourage participants to share both positive and negative experiences with KM, and how you will minimize social desirability bias during interviews.

Response: We appreciate your important suggestions. We have added the following sentences in the ‘Data collection’ section.

(Page 7, Lines 128–132) To minimize social desirability bias and encourage authentic responses, participants will be continuously reminded that sharing challenges or disappointments with treatment is equally important as positive outcomes for the research. Participants will be assured of confidentiality, with explicit guarantees that their responses will remain anonymous and will not be shared with their treating KMDs or any other parties [20].

[20] Tourangeau R, Yan T. Sensitive questions in surveys. Psychological bulletin. 2007;133: 859.

[Comment 7] Risk Management for Emotional Distress:

Since discussing illness experiences may provoke distress, outline whether any support or referral mechanisms will be in place for participants who become upset during interviews.

Response: Thank you for your valuable comments. We have added the following sentences in the ‘Data collection’ section.

(Page 8, Lines 143–145) Recognizing that discussing illness experiences may provoke emotional distress, the research team—which includes healthcare professionals—will monitor participant wellbeing throughout interviews. Participants will be informed of their right to pause or stop the interview at any time.

[Comment 8] Generalizability of Findings:

Briefly discuss the potential limitations regarding the transferability or generalizability of your findings outside the South Korean context.

Response: Thank you for your important suggestions. We have added the potential limitations regarding the generalizability of our findings in the Discussion section as follows.

(Page 10, Lines 206–209) Nevertheless, the generalizability of findings to other contexts may be limited by South Korea’s distinctive dual healthcare system that allows concurrent access to both KM and CM, as well as cultural factors that may influence patient attitudes toward complementary therapies.

[Comment 9] Enhance Readability for International Audience:

Provide brief definitions or explanations for KM modalities (such as pharmacopuncture or moxibustion) for readers who may not be familiar with these terms.

Response: We appreciate your valuable suggestions. We have added brief definitions for KM modalities (pharmacopuncture, thread-embedding acupuncture,

moxibustion, and cupping) in the Introduction section to enhance readability for international readers unfamiliar with these terms.

(Page 3, Lines 32–35) KM offers various modalities for Bell’s palsy, such as acupuncture with electric stimulation, herbal medicine, pharmacopuncture (injection of herbal extracts into acupuncture points), thread-embedding acupuncture (insertion of absorbable threads), moxibustion (burning of mugwort herb on or near the skin), and cupping (suction therapy using cups).

[Comment 10] Minor Language and Formatting:

Carefully proofread the manuscript for minor grammatical, typographical, and formatting errors to improve clarity and readability.

Response: Thank you for your sensitive comments. We have carefully proofread the manuscript and revised the following sentences.

(Page 5, Lines 75–76) Factors such as age, sex, duration of KM treatment, and Bell’s palsy prognosis will be considered to obtain diverse patient experiences.

(Pages 5–6, Lines

---

## [Decision Letter · Decision Letter 1]

28 Aug 2025

Dear Dr. Kim

The proposed study bears significant similarities to the work of Yoon et al. (2022). To further consider this submission, we require the authors to provide a detailed scientific justification that:

1. Clearly identifies the specific gaps or limitations in the Yoon et al. study.

2. Explains how their research aims to address these gaps.

3. Outlines the expected differences in outcomes between their study and the prior work.

We look forward to receiving your revised manuscript.

Kind regards,

Gustav Komlaga

Academic Editor

PLOS ONE

Journal Requirements:

Reviewers' comments:

Reviewer's Responses to Questions

**Comments to the Author**

1. Does the manuscript provide a valid rationale for the proposed study, with clearly identified and justified research questions?

Reviewer #1: Yes

Reviewer #2: Yes

2. Is the protocol technically sound and planned in a manner that will lead to a meaningful outcome and allow testing the stated hypotheses?

Reviewer #1: Yes

Reviewer #2: Yes

3. Is the methodology feasible and described in sufficient detail to allow the work to be replicable?

Reviewer #1: Yes

Reviewer #2: Yes

4. Have the authors described where all data underlying the findings will be made available when the study is complete?

Reviewer #1: Yes

Reviewer #2: Yes

5. Is the manuscript presented in an intelligible fashion and written in standard English?

Reviewer #1: Yes

Reviewer #2: Yes

You may also provide optional suggestions and comments to authors that they might find helpful in planning their study.

Reviewer #1: This is a well-structured and thoughtfully designed qualitative protocol that fills a crucial gap in understanding patient experiences with Korean Medicine for Bell’s palsy. The rationale is compelling, methods are rigorous, and the authors have carefully addressed potential biases, reflexivity, and ethical considerations.

Notably commendable elements include:

- Integration of both purposive and snowball sampling with bias mitigation strategies

- Explicit measures to support participant comfort, authenticity, and emotional safety

- Use of validated qualitative methods and commitment to data saturation

- Clear documentation and transparency in data sharing plans

Minor suggestions:

- Consider proofreading again for small formatting inconsistencies in references or file labeling (e.g., “mMedicine” in one title).

- You may consider expanding dissemination plans for international audiences through multilingual abstracts or summary materials.

I support the acceptance of this study protocol.

Reviewer #2: This revised protocol presents a timely and well-structured qualitative study exploring the experiences of patients with Bell’s palsy undergoing Korean Medicine (KM) treatment. The authors have thoughtfully addressed all prior reviewer comments, resulting in an improved and more transparent protocol.

**Do you want your identity to be public for this peer review?** For information about this choice, including consent withdrawal, please see our Privacy Policy

Reviewer #1: **Yes: ** Shrushti Shah

Reviewer #2: No

---

## [Author Response · Author response to Decision Letter 2]

14 Sep 2025

Responses to the editor's comments

We are pleased to submit a revised version of our manuscript entitled “Experiences of Korean Medicine treatment in patients with Bell’s palsy: a qualitative study protocol” for publication in PLOS ONE.

We appreciate the editor’s comment regarding the similarities to Yoon et al. (2022). We have carefully revised the manuscript to clarify the distinctions between our study and the prior work.

In terms of the research question, Yoon et al. (2022) concentrated on the process of choosing Korean Medicine after Western treatment for facial palsy in general, whereas our protocol focuses on the lived experiences and meanings of Korean Medicine treatment specifically among patients with Bell’s palsy.

From a methodological perspective, the prior study used grounded theory to build a paradigm model of treatment selection, while we apply Colaizzi’s descriptive phenomenological approach to capture the essence of patient experiences.

Concerning participants, the earlier study involved a small sample of ten individuals with mixed etiologies such as cholesteatoma, trauma, Ramsay Hunt syndrome, and central facial palsy, limiting its ability to reflect Bell’s palsy–specific perspectives; our study targets a more homogeneous group of Bell’s palsy patients within two years of onset, with a larger sample until saturation.

Finally, with respect to the expected differences in outcomes, the earlier study highlighted treatment decision-making and suggestions for improving KM, while our work explores motivations, perceptions, and recovery experiences, aiming to inform patient-centered care in Bell’s palsy.

In the revised Introduction, we have added the following sentence:

“Another qualitative study applied grounded theory to focus on the process of choosing KM after CM treatment, but it mixed various etiologies of facial palsy—including cholesteatoma, trauma, Ramsay Hunt syndrome, and central facial palsy—thus lacking a dedicated exploration of Bell’s palsy–specific patient experiences (Yoon et al., 2022)”

We carefully reviewed all references to ensure accuracy and confirm that none of the cited works have been retracted. During this process, we identified one reference (Morrow et al., 2015) with incomplete bibliographic details and therefore removed it from the revised manuscript.

Responses to reviewers' comments

Dear Reviewers,

Thank you for your valuable time and effort in reviewing our manuscript. We believe that the manuscript has been improved based on your thoughtful comments. Changes have been marked as track changes in the manuscript and described with responses.

Reviewer #1

[Comment]

This is a well-structured and thoughtfully designed qualitative protocol that fills a crucial gap in understanding patient experiences with Korean Medicine for Bell’s palsy. The rationale is compelling, methods are rigorous, and the authors have carefully addressed potential biases, reflexivity, and ethical considerations.

Notably commendable elements include:

- Integration of both purposive and snowball sampling with bias mitigation strategies

- Explicit measures to support participant comfort, authenticity, and emotional safety

- Use of validated qualitative methods and commitment to data saturation

- Clear documentation and transparency in data sharing plans

Minor suggestions:

- Consider proofreading again for small formatting inconsistencies in references or file labeling (e.g., “mMedicine” in one title).

- You may consider expanding dissemination plans for international audiences through multilingual abstracts or summary materials.

I support the acceptance of this study protocol.

Response:

We sincerely thank the reviewer for the generous and encouraging evaluation of our study protocol. Regarding the minor suggestions, we carefully proofread the references and corrected formatting inconsistencies. In the revised manuscript, we modified the Dissemination section as follows:

“We intend to disseminate the findings of this study through academic journals, seminars, and conference presentations, including the preparation of multilingual abstracts to reach international audiences.”

We believe this addition will clarify our plan to make the study findings more accessible beyond domestic contexts.

Reviewer #2

[Comment]

This revised protocol presents a timely and well-structured qualitative study exploring the experiences of patients with Bell’s palsy undergoing Korean Medicine (KM) treatment. The authors have thoughtfully addressed all prior reviewer comments, resulting in an improved and more transparent protocol.

Response:

We sincerely appreciate the reviewer’s positive evaluation and encouragement.

---

## [Editor Report · Decision Letter 2]

15 Sep 2025

Experiences of Korean Medicine treatment in patients with Bell’s palsy: a qualitative study protocol

PONE-D-24-46812R2

Dear Dr. Kim

We’re pleased to inform you that your manuscript has been judged scientifically suitable for publication and will be formally accepted for publication once it meets all outstanding technical requirements.

Kind regards,

Gustav Komlaga

Academic Editor

PLOS ONE
---

## [Editor Report · Acceptance letter]

PONE-D-24-46812R2

PLOS ONE

Dear Dr. Kim,

I'm pleased to inform you that your manuscript has been deemed suitable for publication in PLOS ONE. Congratulations! Your manuscript is now being handed over to our production team.

Kind regards,

on behalf of

Prof. Gustav Komlaga

Academic Editor

PLOS ONE